# Pre-Treatment and Preoperative Neutrophil-to-Lymphocyte Ratio Predicts Prognostic Value of Glioblastoma: A Meta-Analysis

**DOI:** 10.3390/brainsci12050675

**Published:** 2022-05-21

**Authors:** Xin Guo, Hengxing Jiao, Tiantian Zhang, Yuelin Zhang

**Affiliations:** 1Department of Graduate Work, Hanguang Campus of Xi’an Medical University, Xi’an 710068, China; gx1217449964@163.com (X.G.); jiaohengxingfx@163.com (H.J.); nicole06080520@163.com (T.Z.); 2Vice President of Xi’an Medical University, Weiyang Campus of Xi’an Medical University, Xi’an 710068, China

**Keywords:** glioblastoma, GBM, glioblastoma multiforme, NLR, neutrophil-to-lymphocyte ratio, prognostic

## Abstract

Objective: Although some meta-analyses have shown a correlation between a high neutrophil-to-lymphocyte ratio (NLR) and low survival in patients with gliomas, their conclusions are controversial, and no study has specifically explored the relationship between a high pre-treatment and pre-operative NLR and low survival in patients with glioblastoma (GBM). Therefore, we further investigated this correlation through meta-analysis. Methods: We searched the PubMed, Metstr, and Cochrane databases in March 2022 for published literature related to high pre-treatment and pre-operative NLR and low survival in patients with GBM. The literature was rigorously searched according to inclusion and exclusion criteria to calculate the overall hazard ratio (HR) and 95% confidence interval (CI) corresponding to a high NLR using a random effects model. Results: The total HR for the pre-treatment and pre-operative NLR was 1.46 (95% CI: 1.17–1.75, *p* = 0.000, I^2^ = 76.5%), indicating a significant association between a high pre-treatment and pre-operative NLR, and low overall survival in patients with GBM. Sub-group analysis was performed because of the high heterogeneity. The results for the sub-group with a cut-off value of 4 showed an HR of 1.39 (95% CI: 1.12–1.65, *p* = 0.000, I^2^ = 22.2%), with significantly low heterogeneity, whereas those for the sub-group without a cut-off value of 4 showed an HR of 1.45 (95% CI: 1.01–1.89, *p* = 0.000, I^2^ = 83.3%). Conclusions: The results of this study demonstrate that a high pre-treatment and pre-operative NLR suggests low survival in patients with GBM based on data from a large sample. Furthermore, the meta-regression analysis results indicate that underlying data, such as age and extent of surgical resection, lead to a high degree of heterogeneity, providing a theoretical basis for further research.

## 1. Introduction

Glioblastoma (GBM) is a common, highly malignant glioma (grade IV) that is often accompanied by high morbidity and mortality, with a 5-year overall survival (OS) rate of <5% [1]. Despite a range of standardised treatments, such as surgery combined with radiotherapy (e.g., temozolomide) [2], patient factors and tumour heterogeneity have led to significant differences in prognostic outcomes [3]. In recent years, several studies have demonstrated the prognostic significance of the peripheral blood neutrophil-to-lymphocyte ratio (NLR) in glioblastoma [4,5] and other cancers [6,7,8]. The NLR is a convenient, low-cost, and easily measured marker that responds to the balance between inflammation and the immune system. Numerous studies have shown that increased lymphocytes around the tumour or in the peripheral blood often suggest a poor prognosis, whereas lymphocytosis indicates a better prognosis [9,10,11,12,13,14].

Several meta-analyses have shown that a high NLR is associated with poor survival in patients with glioma [15,16,17], and one [18] reported a correlation between high NLR and glioma grade. However, no meta-analysis has been performed on studies with a high NLR and poor overall survival (OS) prior to GBM treatment and surgery. In the studies by Lei et al. [15] and Yang et al. [17], sub-group analyses showed a correlation between a high NLR and poor OS in patients with GBM. However, these analyses only indicated a small number of studies. Furthermore, studies by He et al. [19] and Besiroglu et al. [20] demonstrated that a high NLR was not associated with poor OS in patients with GBM. Since only high-grade gliomas and patients who underwent standardised surgery were included in the study by He et al. [19], confounding factors for low-grade gliomas were excluded, restricting these findings to a smaller subset of GBM patients. Besiroglu et al. [20] reported that an increase in pre-treatment neutrophil count might have improved the treatment response to bevacizumab in patients with GBM, leading to an inability to demonstrate the prognostic significance of NLR. However, this needs to be validated by further studies. Given the differences in correlations between a high NLR and poor OS before GBM treatment, closer examination of this relationship is warranted.

## 2. Methods

### 2.1. Identification of Studies and Collection of Data

We conducted this meta-analysis based on the recommendations and criteria developed by the Preferred Reporting Items for Systematic Reviews and Meta-Analyses [21]. A search was conducted in March 2022 to identify all literature on pre-treatment and pre-procedure NLR related to GBM survival. Searches were performed using the terms “glioblastoma” or “GBM” or “glioblastoma multiforme” and “NLR” or “neutrophil/lymphocyte ratio” and “prognosis” and their variants in the PubMed, Metstr (http://fmrs.metstr.com/index.aspx (accessed on 5 March 2022 )), and Cochrane databases. The final literature for inclusion was screened manually by two researchers (X.G. and H.J.) and reviewed by a third researcher (T.Z.) to avoid omissions. Finally, basic data were extracted from the included literature, including information on the first author, country, year of publication, type of study, age, sex, sample size, extent of surgical resection, number of patients undergoing secondary surgery, NLR thresholds, and treatment options (Table 1). In addition, we extracted corresponding survival rates and 95% confidence interval (CI) values. In cases where both univariate and multivariate analysis results were available, we prioritised extraction of outcome values from multivariate analysis. For the literature where hazard ratio (HR) values and 95% CIs for the corresponding OS were not mentioned, but the corresponding Kaplan–Meier curves were provided, survival data were extracted and converted to HR and 95% CIs using Engauge Digitiser version 12.1 (Xi’an, China) with the interrupted point-taking method.

### 2.2. Inclusion and Exclusion Criteria

Adherence to the inclusion and exclusion criteria helped avoid the influence of surgery and radiotherapy on screening results. Literature addressing the prognostic value of the pre-treatment and pre-operative NLR in patients with GBM and including HR values for OS and their corresponding 95% CIs or Kaplan–Meier curves as well as the number of patients were included. Literature with unclear sources of the NLR, measuring NLR values extracted during radiotherapy, and including patients with chronic diseases or cachexia, which affect NLR values; basic experiments, similar meta-analyses, reviews, letters, and duplicates from different sources were excluded.

### 2.3. Quality Assessment

Two independent reviewers (X.G. and H.J.) used the Newcastle-Ottawa Quality Assessment Scale (NOS) [22] to assess the quality of the included studies. The scale consists of three main items—selection (0–4 points), comparability control for important factors (0–2 points), and outcome (0–3 points). Each main item also includes sub-items with a total score of 9 (Table 2). Based on the relevant literature, studies with an NOS score ≥ 5 were considered high-quality studies [22]. In addition, the senior author (Y.Z.) resolved any disputes between the two reviewers regarding results.

### 2.4. Statistical Analysis

All meta-data analyses were performed using Stata version 16.0 (Xi’an, China). The overall HRs and 95% CIs for the pre-treatment and pre-operative NLR values from the literature were calculated and presented as forest plots. Null was defined as the point when the value for the vertical axis of the plot exceeded 1, that is, when the combined HR and 95% CI surpassed 1, suggesting that a high NLR is not associated with low OS in patients with GBM (*p* < 0.05). Second, the Higgins I-squared (I^2^) statistic was used to assess the heterogeneity among studies. Following the Cochrane Handbook criteria, I^2^ > 50% indicated significant heterogeneity [23]. In these cases, a random effects model was chosen to analyse the final results. Otherwise, a fixed effects model was selected. In addition, when there was significant heterogeneity, we performed sub-group, meta-regression, and sensitivity analyses to search for sources of heterogeneity and increase result robustness. Finally, we used funnel plots to detect publication bias. If asymmetry was present in the funnel plots, Egger’s linear regression and Begg’s rank correlation were used to quantitatively evaluate whether there was a publication bias.

## 3. Results

### 3.1. Search Results

Among the 106 studies obtained based on the search criteria, 87 were excluded for the following reasons: post-operative NLR and unclear NLR origin (*n* = 51), incorrect markers (*n* = 6), patients did not fulfil the inclusion criteria (*n* = 22), letters (*n* = 3), and meta-analysis (*n* = 5) (Figure 1). For the final analysis, 19 studies were identified [4,9,19,20,24,25,26,27,28,29,30,31,32,33,34,35,36,37,38], and the main characteristics of the included studies are shown in Table 1. A total of 2762 patients (938 men and 1824 women) were enrolled in 19 studies. The studies were from China (*n* = 9), Turkey (*n* = 4), the USA (*n* = 1), Portugal (*n* = 1), the Netherlands (*n* = 1), Israel (*n* = 1), Italy (*n* = 1), and France (*n* = 1). The extent of surgical resection was also assessed, such as gross total resection (GTR) in 789 patients and other resection measures in 1973 patients. Notably, only one study reported 23 patients who underwent a second surgery, and one study reported the pre-treatment NLR. In addition, NLR thresholds varied in the included literature, with 10 studies having a threshold value of 4 and the remaining 9 studies with thresholds of 2.39, 7, 2.7, 4.1, 2.9, 2.42, 5, 7.25, and 3.31. Seven studies had a post-operative treatment regimen of Stupp + chemoradiotherapy + temozolomide, 3 studies had a post-operative treatment regimen of Stupp + chemoradiotherapy, and 9 studies did not report a post-operative treatment regimen. In terms of analytical methods, 5 studies were obtained by univariate analysis, 2 by unknown analytical methods, and 12 by multivariate analyses.

### 3.2. Correlation between OS and NLR in GBM

The final HR value and corresponding 95% CI was 1.46 (1.17–1.75, *p* = 0.000), calculated using Stata version 16.0, indicating a significant correlation between high values of the pre-treatment NLR and low OS in patients with GBM (Figure 2, Tau-squared = 0.2504). Heterogeneity analysis revealed an I^2^ value of 76.5% (*p* = 0.000), indicating significant heterogeneity. To clarify whether the heterogeneity was due to studies involving secondary surgery, the overall HR for the NLR was calculated; it was found to be 1.51 (95% CI: 1.2–1.8, *p* = 0.000, I^2^ = 77.8%) after removing a study on secondary surgery (Appendix A). Subsequently, we performed sub-group analysis based on the presence or absence of a cut-off value of 4. The HR for the sub-group with a cut-off value of 4 was 1.39 (95% CI: 1.12–1.65, *p* = 0.000, I^2^ = 22.2%), showing a significant decrease in heterogeneity, indicating that a preoperative NLR of >4 suggests a low OS rate in patients with GBM. In contrast, the HR for the sub-group with a cut-off value other than 4 was 1.45 (95% CI: 1.01–1.89, *p* = 0.000, I^2^ = 83.3%) (Figure 3), showing a high level of heterogeneity. However, further sub-group analysis could not be performed to find the source of heterogeneity because of the different cut-off values for each sub-group. The subsequent sensitivity analysis for the sub-group with a cut-off value other than 4 showed that individual studies were not associated with heterogeneity (Appendix A).

### 3.3. Heterogeneity and Sensitivity Analysis

To further identify sources of heterogeneity, sensitivity analysis was performed, revealing that single-factor studies were not associated with high heterogeneity (Figure 4). Sub-group analysis was again performed for univariate, multivariate, and unknown sources, which showed HRs of 1.41 (95% CI: 0.68–2.14, *p* = 0.000, I^2^ = 86.5%), 1.45 (95% CI: 1.09–1.81, *p* = 0.000, I^2^ = 69.6%), and 1.80 (95% CI: 1.19–2.41, *p* = 0.000, I^2^ = 0.0%), respectively. Sub-group analysis indicated that the heterogeneity in both univariate and multivariate sources was not significantly reduced (Figure 5). Finally, we performed meta-regression analysis to evaluate whether country, year of publication, age, sex, extent of surgical resection, and treatment method affected the heterogeneity. The results revealed a heterogeneity of Tau-squared = 0 after inclusion of these six variables, which was 0.2504, less than the previous value of 0.2504 (Figure 2). These data imply that country, year of publication, age, sex, extent of surgical resection, and treatment method can be used to explain 76.5% heterogeneity among studies (Table 3).

### 3.4. Publication Bias

Since 19 studies were included, we used funnel plots to check for the presence of publication bias. Egger’s and Begg’s tests were used to quantitatively assess the presence of publication bias after finding funnel plot asymmetry (Appendix B Figure A1, Figure A2 and Figure A3). As shown in Table 4, the *p*-value obtained was 0.066 > 0.05 (95% CI: −0.3302535–9.108586), indicating that there was no publication bias in the included studies.

## 4. Discussions

The prognostic significance of a high NLR before GBM treatment and surgery was evaluated in 19 studies, including 2762 patients. The results of a parallel meta-analysis of data from published literature found that high pre-treatment and preoperative NLRs indicate poor and statistically significant survival in patients with GBM.

Numerous studies have shown that the NLR is associated with the prognosis of certain malignancies [39,40], but a significant prognostic value has not been found in intermediate diseases, such as breast cancer [41] and gastric cancer [42]. Although some studies have shown that the preoperative NLR is associated with poor OS in patients with GBM, Weng et al. [23] and Zhou et al. [36] reported conflicting results. The results reported by Brenner et al. [29] and Bambury et al. [4] also differed, although the NLR thresholds and treatments were identical and multivariate analyses were performed. To date, the mechanism underlying the prognostic role of the NLR in GBM is unclear [19], but one study has proposed that in the GBM microenvironment, tumour-infiltrating lymphocytes are the main regulatory T cells capable of suppressing the immune response [43]. Another study suggested that the NLR may drive cancer cell proliferation by increasing the availability of growth factors, angiogenic factors, and other pro-neoplastic signalling molecules, thus portending a worse prognosis [44,45].

Although previous meta-analyses have shown a correlation between high NLR values and poor OS in patients with GBM, the result was based on sub-group analysis that included not only a small body of literature but also different sources of the NLR. Furthermore, the meta-analysis by Wang et al. [15,16,17] included the study by Mason et al. [46]. The application of hormone therapy, radiotherapy, and chemotherapy can affect NLR values [47]. Additionally, in the meta-analysis of Wang et al. [16], the included studies by Weng et al. [25] and Lei et al. [15] contained data inclusion errors, suggesting publication bias. However, the quantitative results were not assessed using tests, such as Egger’s test. We believe that these issues ultimately affect the credibility of the results. Therefore, we expanded our sample size to avoid these problems and investigated the correlation between high pre-treatment NLR values and poor OS in patients with GBM. Using pre-treatment NLR values as the study index and excluding NLR literature obtained during the application of hormone therapy or radiotherapy, we showed that high pre-treatment NLR values suggested poor prognosis in patients with GBM (HR = 1.46; 95% CI: 1.17–1.75, *p* = 0.000, I^2^ = 76.5%). Owing to the presence of high heterogeneity, we performed sub-group analysis of an NLR with different cut-off values and found significantly lower heterogeneity in sub-group analysis of the NLR with a cut-off value of 4 (I^2^ = 22.2%) and without a cut-off value of 4 (I^2^ = 83.3%). However, sensitivity analysis failed to clarify the source of high heterogeneity. This result is consistent with the findings reported by Bambury et al. [4,5], who demonstrated that an NLR > 4 is an independent indicator of poor prognosis in patients with GBM.

Among the included studies, one study reported 23 patients who underwent a second surgery [31]. Secondary surgery can significantly prolong the survival of patients with recurrent GBM [48]. This may be attributed to the ability of secondary surgery to overcome the negative impact of the first incomplete resection on survival [49] and reduce the residual tumour volume [50]. However, studies have confirmed that the area of repeat craniectomy does not affect survival in patients who achieve the GTR value with the first surgery, whereas survival is significantly improved if the first surgery involves subtotal resection [49]. The effect of secondary surgery on patient survival is controversial and requires further investigation.

A high pre-operative NLR allows for glioma grading [23]; this may be related to the progressive features of gliomas, such as inflammation, angiogenesis, and invasion. Neutrophils are the first to reach the site of inflammation [51], and as the tumour grade increases, elevated neutrophil levels inhibit the cytolytic activity of quiescent lymphocytes, leading to a decrease in the number of lymphocytes or reduced function [52]. In addition, glioma-derived factors may affect the number of circulating and infiltrating neutrophils, whilst promoting GBM cell proliferation by upregulating S100A4 [13,53] and thereby affecting their infiltration into tumours [14]. However, studies have also shown that the specificity of neutrophils is low and increases in the presence of subclinical infections and acute stress [23]. Additional studies are required to confirm their role in tumours.

Furthermore, a high pre-operative NLR can distinguish between patients with non-lesional epilepsy, acoustic neuroma, and meningioma; healthy individuals; and patients with glioma. It also has the highest diagnostic accuracy for glioma not only as an indicator for determining glioma grading but also for identifying isocitrate dehydrogenase wild-type and mutant GBM [54,55]. Although the NLR cannot distinguish between GBM and metastases [56] as an inflammatory marker of the host, it can provide, to some extent, useful information for clinical treatment options [5]. For example, as the NLR increases, tumour aggressiveness and poor prognosis also increase, for which aggressive treatment modalities (such as targeted therapy or immunotherapy) are therapeutic options [34]. In addition, anti-neutrophil factor drugs can be used to treat an elevated NLR [54]. Moreover, an increase in the pre-operative NLR can guide the periodic review of patients for the assessment of tumour progression [57].

Although we included, for the first time, 19 studies exploring the relationship between the NLR and GBM prognosis with optimistic results, this study has some limitations. First, the critical values of the NLR in the included literature vary, and it is not possible to compare between the critical values and define a critical value for a high NLR, making the clinical judgement of a high pre-operative NLR controversial. Second, one of the included studies reported secondary surgery, which may prolong patient survival. Third, most included studies had a retrospective design, and although we performed meta-regression to identify sources of heterogeneity, we were unable to find prospective clinical studies. A large multicentre prospective randomised study to determine the correlation between high a pre-treatment and pre-operative NLR and poor OS in patients with GBM is warranted to increase the validity of our findings.

## 5. Conclusions

For the first time, this study demonstrated a correlation between a high pre-treatment and pre-operative NLR and poor OS in patients with GBM using a large sample size. The study results demonstrated that a high NLR suggests reduced survival in patients with GBM, providing a theoretical basis for prospective clinical studies. Although the study results showed a high degree of heterogeneity, underlying data on age, sex, and extent of surgical resection were identified as sources of heterogeneity after meta-regression analysis. A high NLR may be used to predict the survival of patients with GBM and can serve as a basis for adopting appropriate interventions to improve their prognosis.

## Figures and Tables

**Figure 1 brainsci-12-00675-f001:**
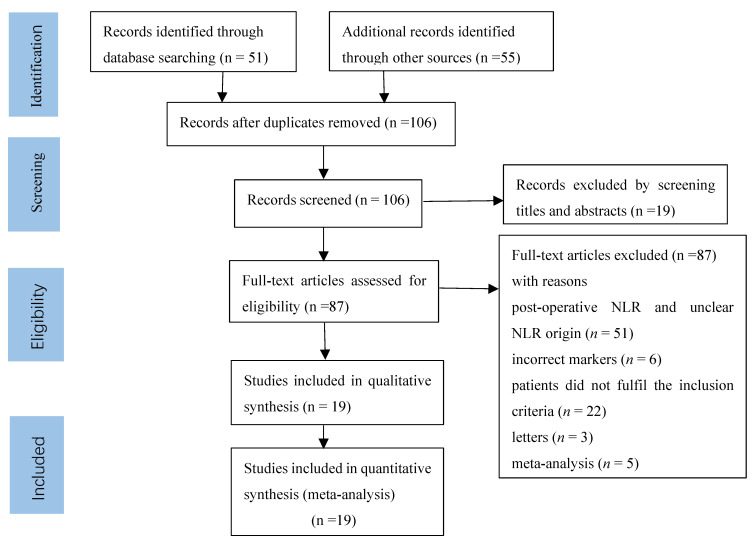
Selection process for the including studies.

**Figure 2 brainsci-12-00675-f002:**
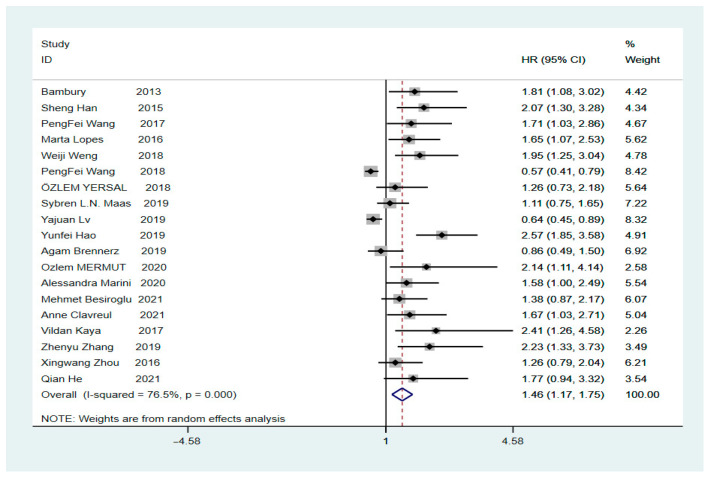
Forest plot illustrating the relationship between the NLR and OS in glioblastoma patients [4,9,19,20,22,23,24,25,26,27,28,29,30,31,32,33,34,35,36].

**Figure 3 brainsci-12-00675-f003:**
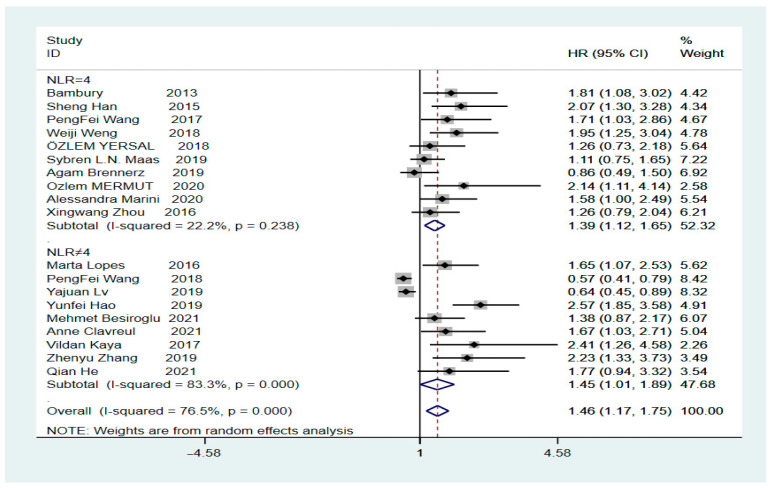
Inclusion of subgroups with a literature cut-off value of 4 and subgroups without a cut-off value of 4 in the analysis [4,9,19,20,22,23,24,25,26,27,28,29,30,31,32,33,34,35,36].

**Figure 4 brainsci-12-00675-f004:**
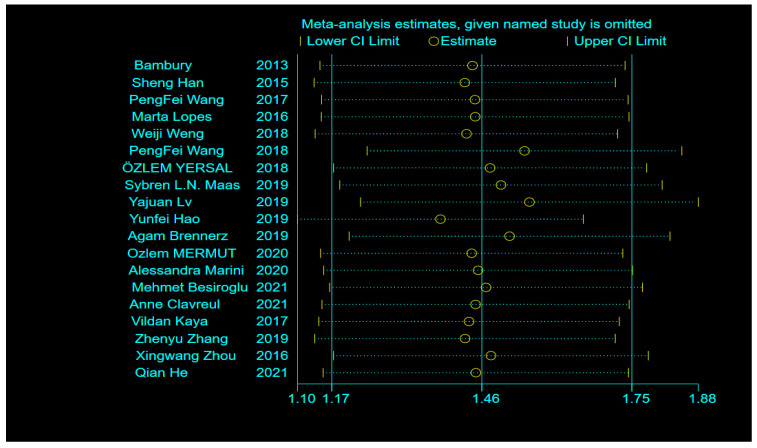
Sensitivity analysis of the relationship between single studies and heterogeneity in the included literature [4,9,19,20,22,23,24,25,26,27,28,29,30,31,32,33,34,35,36].

**Figure 5 brainsci-12-00675-f005:**
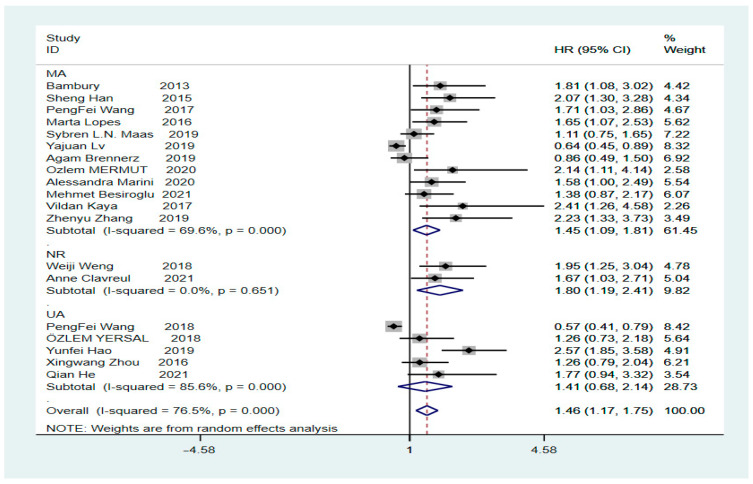
Analysis of subgroups with different sources of endings [4,9,19,20,22,23,24,25,26,27,28,29,30,31,32,33,34,35].

**Table 1 brainsci-12-00675-t001:** Main characteristics of 19 studies included in meta-analysis.

First Author and Year	Country	Patients (N)	Design	Age (mean ± SD)	Gender (M)	Sample Size (N)	Gross Total Resection (N)	Second Surgery (N)	NLR Cut-off	Therapy	Outcome	HR (95%CI)
Bambury 2013	USA	GBM multiforme	Retrospective	56.6 ± 11.53	65	84	23	NR	NLR = 4	S + C+ T	MA	1.81 (1.08–3.02)
Sheng Han 2015	China	GBM	Retrospective	50.4 ± 15.4	95	152	75	NR	NLR = 4	S + C	MA	2.068 (1.304–3.277)
Peng-Fei Wang 2017	China	GBM	Prospective	52.1 ± 0.984	96	166	102	NR	NLR = 4	NR	MA	1.712 (1.026–2.858)
Marta Lopes 2018	Portugal	GBM multiforme	Retrospective	NR	NR	126	NR	NR	NLR = 5	NR	MA	1.56 (1.04–2.34)
Weiji Weng 2018	China	GBM	Retrospective	NR	53	105	57	NR	NLR = 4	S + C+ T	NR	1.953 (1.255–3.039)
Peng-Fei Wang 2018	China	GBM	Retrospective	NR	NR	314	NR	NR	NR	NR	UA	0.57 (0.41–0.79)
ÖZLEM YERSAL 2018	Turkey	GBM	Retrospective	56.8 ± 13.1	39	80	42	NR	NLR = 4	S + C+ T	UA	1.258 (0.727–2.179)
SybrenL.N. Maas 2019	Netherlands	GBM	Retrospective	NR	NR	479	NR	NR	NLR = 4	NR	MA	1.11 (0.75–1.65)
Yajuan Lv 2019	China	GBM	Retrospective	53.25 ± 13.9	113	192	NR	NR	NLR = 2.7	NR	MA	0.637 (0.454–0.894)
Yunfei Hao 2019	China	GBM multiforme	Retrospective	55 ± 13.55	116	187	112	NR	NLR = 4.1	S + C+ T	UA	2.574 (1.849–3.581)
Agam Brenner 2019	Israel	GBM multiforme	Retrospective	57.73 ± 12.43	46	89	59	23	NLR = 4	S + C+ T	MA	0.856 (0.49–1.496)
Ozlem MERMUT 2020	Turkey	GBM multiforme	Retrospective	58.0 ± 13.02	47	75	31	NR	NLR = 4	NR	MA	2.14 (1.11–4.14)
Alessandra Marini 2020	Italy	GBM	Retrospective	NR	65	124	64	NR	NLR = 4	S + C	MA	1.58 (1–2.49)
Mehmet Besiroglu 2021	Turkey	GBM multiforme	Retrospective	47.2 ± 12.1	58	107	73	NR	NLR = 2.9	S + C+ T	MA	1.38 (0.87–2.17)
Anne Clavreul 2021	France	GBM	Retrospective	61.5 ± 8.8	65	85	46	NR	NLR = 2.06	S + C+ T	NR	1.67 (1.03–2.71)
Vildan Kaya 2017	Antalya	GBM	Retrospective	55.7 ± 16.3	NR	90	NR	NR	NLR = 5	NR	MA	2.41 (1.26–4.58)
Zhen-Yu Zhang 2019	China	GBM	Retrospective	NR	NR	170	NR	NR	NLR = 7.25	NR	MA	2.228 (1.329–3.733)
Xing-Wang Zhou 2016	China	GBM	Retrospective	52.85 ± 4.03	50	84	59	NR	NLR = 4	C	UA	1.264 (0.785–2.035)
Qian He 2021	China	GBM	Retrospective	NR	NR	62	46	NR	NLR = 3.31	S + C	UA	1.766 (0.941–3.316)

Note: UA: univariate analysis; MA: multivariate analysis; S + C + T: Stupp + chemoradiotherapy + temozolomide; NLR: neutrophil count/lymphocyte count; HR: hazard ratio; CI: confidence interval; NR: not reported; NOS: Newcastle-Ottawa quality assessment.

**Table 2 brainsci-12-00675-t002:** Results of quality assessment using the Newcastle-Ottawa Scale for case-control studies.

Author and Year	Selection (0–4 points)				Comparability Control for Important Factor (0–2 points)	Outcome (0–3 points)			Scores (9 Points)
	Representativeness of the Exposed Cohort	Selection of the Nonexposed Cohort	Ascertainment of Exposure	Demonstration that Outcome of Interest was not Present at Start of Study		Assessment of Outcome	was Follow-Up Long Enough for Outcomes to Occur	Adequacy of Follow-Up of Cohorts	
Bambury 2013	1	1	1	1	1	1	1	1	8
Sheng Han 2015	1	1	1	1	1	0	1	0	6
Peng-Fei Wang 2017	1	1	1	1	1	1	1	0	7
Marta Lopes 2018	1	1	1	1	0	0	1	0	5
Weiji Weng 2018	1	1	0	1	1	1	0	0	5
Peng-Fei Wang 2018	1	1	0	1	1	1	0	1	6
ÖZLEM YERSAL 2018	1	0	1	1	1	1	0	0	5
SybrenL.N. Maas 2019	1	1	1	1	1	1	1	1	8
Yajuan Lv 2019	1	1	1	1	1	1	0	1	7
Yunfei Hao 2019	1	1	1	1	0	1	1	0	6
Agam Brenner 2019	1	1	1	1	1	1	1	0	7
Ozlem MERMUT 2020	1	1	1	1	1	1	1	1	8
Alessandra Marini 2020	1	1	1	1	0	1	0	0	5
Mehmet Besiroglu 2021	1	1	0	1	1	1	1	0	6
Anne Clavreul 2021	1	1	1	1	1	1	0	1	7
Vildan Kaya 2017	1	1	0	1	1	1	0	0	5
Zhen-Yu Zhang 2019	1	1	0	1	1	1	1	1	7
Xing-Wang Zhou 2016	1	0	1	1	1	1	1	1	7
Qian He 2021	1	1	1	0	1	1	1	1	7

**Table 3 brainsci-12-00675-t003:** Meta-regression of baseline data.

Hr	Coef.	Std. Err.	t	*p* > |t|	[95% Conf.]	Interval
year	−0.0623897	0.2007026	−0.31	0.761	−0.4996832	0.3749038
country	0.0193201	0.1760801	−0.11	0.914	−0.4029657	0.3643255
age	0.1452009	0.2485765	0.58	0.57	−0.3964008	0.6868027
gender	0.879301	0.1934452	0.45	0.658	−0.3335507	0.5094109
Extent of surgery	−0.1954218	0.3527367	−0.55	0.59	−0.963969	0.5731255
treatment	0.1881377	0.4750795	0.4	0.699	−0.8469717	1.223247
cons	1.393349	2.010993	0.69	0.502	−2.988229	5.774926

Note: Meta-regression Number of obs = 19; REML estimate of between-study variance tau2 = 0; % residual variation due to heterogeneity I-squared_res = 0.00%; Proportion of between-study variance explained Adj R-squared = %; Joint test for all covariates Model F (6,12) = 0.25; With Knapp–Hartung modification Prob > F = 0.9513.

**Table 4 brainsci-12-00675-t004:** Egger’s test.

Std_Eff	Coef.	Std. Err	t	P > |t|	[95% Conf. Interval]	
Slope	−0.6745632	0.5180683	−1.30	0.210	−1.767592	0.4184653
Bias	4.389166	2.236887	1.96	0.066	−0.3302535	9.108586

## Data Availability

Not applicable.

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
