# Peer review of "Pre-Treatment and Preoperative Neutrophil-to-Lymphocyte Ratio Predicts Prognostic Value of Glioblastoma: A Meta-Analysis"

_brainsci, 2022, doi:10.3390/brainsci12050675_

Round 1

Reviewer 1 Report

In this study, the author analyzed the correlation between preoperative NLR and adverse OS by summarizing the data from the published literature. Generally, The results of a parallel meta-analysis of data from the qualified published literature found that high preoperative NLR was associated with poor OS in GBM and was statistically significant. However, by combining and reanalyzing the published data, the authors just solidify the conclusion that NLR could be a potential marker for the poor OS, which was already reported by several studies before and the data collected by the authors were unable to support the authors to do further analysis, like the analysis of the sex and age. So this study lacked novelty and the conclusion had limited contribution to the field.

Author Response

Dear Reviewer

        Hello

        I am Xin Guo, thank you for your valuable suggestions and comments; I will answer your questions one by one and suggest corrections in the draft response letter. we rewrote most of the introduction to fully describe the methodology and further explicitly show the results, and rewrote the conclusions.

        Again, we found that although previous meta-analyses showed a correlation between high NLR and poor OS in glioma, the role of high NLR in glioma grading is controversial, and even if the meta-analysis showed a relationship between high NLR and glioma survival, it does not mean that high NLR is equally present in GBM prognosis. Although there were also subgroup analyses showing a correlation between high NLR and poor OS in GBM, fewer included literature, and their studies included literature on NLR extracted during radiotherapy, which generally affects NLR values. In addition, we conducted a first analysis of the sources of heterogeneity using meta-regression after transforming the data by age and gender and found that the underlying data such as age and gender were the main sources of heterogeneity. We also discuss the role of secondary surgery on glioma prognosis and the high NLR values in glioma grading, which were not available in previous studies. Finally, these changes will be reflected in the draft of the reply letter one by one. Thank you again for your valuable comments and I sincerely wish you good luck and good health.

          Finally, thank you again for your valuable comments, and I sincerely wish you good work and good health.

Yours sincerely

Salutations

Reviewer 2 Report

The present article entitled "Preoperative neutrophil-to-lymphocyte ratio predicts prognostic value of glioblastoma: A meta-analysis" investigated the correlation between NLR and poor OS in GBM focusing of 20 studies. 

The introduction and methods are sufficient described but should be improved.

The authors should improve the english language and style, because several sentences need to be revised and edited.

Author Response

Dear Reviewer

      Hello

      I am Xin Guo, first of all, thank you for your valuable suggestions and comments. I will answer your questions one by one and suggest corrections in the draft response letter.

Question1 The introduction and methods are sufficient described but should be improved.

       We have rewritten the introduction and methods section

Question2 The authors should improve the english language and style, because several sentences need to be revised and edited.

        We enlisted a professional English editing service to do the English touch-ups; these changes will be reflected in the returned manuscript.

        Finally, thank you again for your valuable comments, and I sincerely wish you good work and good health.

Yours sincerely

Salutations

Reviewer 3 Report

This topic is very current, but paper needs some revisions. Please look at these point to improve the manuscript:

  • Literature search was well done, statistical results are good in the text, however in the abstract is very difficult to understand what this meta-analisis add to the literature. Please revise and improve.
  • "Numerous studies have shown that peritumor or peripheral blood neutrophilia tends to suggest a poor prognosis.. " it is reported numerous, but just 2 references appear. Please add more refs, look at: --  doi: 10.1007/s13760-021-01765-4.  -- doi: 10.1016/j.semcancer.2013.02.001
  • "7studies were glioblastoma multiforme and the remaining 13 studies were glioblastoma" What did authors want to say with this sentence?
  • In table 1, it seems that among the 20 included studies, none of them reported second surgery. Is it correct? The role of second surgery in recurrent GBM and its impact on prognosis should be discuss in the discussion section. Look at these papers: -- doi: 10.1016/j.clineuro.2021.106735 -- doi: 10.1007/s00066-021-01884-0
  • Were blood samples taken from all patients prior to surgery? Is it always possible to define these data from all studies? if not, this could be a limitation of the review and should be reported.
  • A similar paper (Gomes Dos Santos et al. Role of neutrophil-lymphocyte ratio as a predictive factor of glioma tumor grade: A systematic review. Crit Rev Oncol Hematol. 2021 Jul;163:103372 ) demonstrated that patients with high NLR values were diagnosed with high-grade gliomas. This point should be discuss in the discussion section.

Author Response

Dear Reviewer

        Hello

        I am Xin Guo, first of all, thank you for your valuable suggestions and comments; secondly, I will answer your questions one by one and make changes one by one in the draft reply.

Question1 Literature search was well done, statistical results are good in the text, however in the abstract is very difficult to understand what this meta-analisis add to the literature. Please revise and improve.

          Our study, while further expanding the sample size, demonstrates the low survival rate of GBM before  high Pre-treatment and at preoperative NLR . In addition, we used meta-regression for the first time to demonstrate that age, year of publication, country, and extent of surgical resection were sources of heterogeneity in these underlying data, whereas meta-regression analysis of sources of heterogeneity was not available in previous studies, facilitating a theoretical basis for subsequent studies.

Question2 Numerous studies have shown that peritumor or peripheral blood neutrophilia tends to suggest a poor prognosis. " it is reported numerous, but just 2 references appear. Please add more refs.

           We have added several references by reviewing the literature, which can be clearly seen in the returned manuscript.

Question3 7studies were glioblastoma multiforme and the remaining 13 studies were glioblastoma" What did authors want to say with this sentence?

           This is a misrepresentation by the author and has been corrected and submitted

Question4 In table 1, it seems that among the 20 included studies, none of them reported second surgery. Is it correct? The role of second surgery in recurrent GBM and its impact on prognosis should be discuss in the discussion section.

            The authors again reviewed the included literature and found only one reference mentioning 23 patients who underwent secondary surgery [DOI:10.1159/000500926], subsequently we discuss the role of secondary surgery in GBM i.e. and prognostic impact in the discussion section, which will be reflected in the return manuscript.

Question5 Were blood samples taken from all patients prior to surgery? Is it always possible to define these data from all studies? if not, this could be a limitation of the review and should be reported.

             By reviewing the literature again, we found one study that was not a preoperative study [DOI:10.1016/j. jocn.2021.01.036], which was subsequently excluded for reasons of article rigor. In addition, one study did not specify whether surgery was performed and the extracted pre-treatment NLR values [DOI:10.22034/APJCP.2017.18.12.3287], and we considered these literature for inclusion and changed the study objective to the prognostic significance of pre-treatment and preoperative NLR in GBM to make the article more rigorous

Question6 A similar paper (Gomes Dos Santos et al. Role of neutrophil-lymphocyte ratio as a predictive factor of glioma tumor grade: A systematic review. Crit Rev Oncol Hematol. 2021 Jul; 163: 103372) demonstrated that patients with high NLR values were diagnosed with high-grade gliomas. This point should be discuss in the discussion section.

            By reading the literature, we present the graded role of NLR in glioma in the discussion section.

            Finally, thank you again for your valuable comments, and I sincerely wish you good work and good health.

Yours sincerely

Salutations

Round 2

Reviewer 1 Report

The authors made extensive revisions and re-organized the manuscript. The authors clearly demonstrate the purpose, novelty, and results of the study in the new version of the manuscripts. The quality of the new manuscript was greatly improved. However, there are still some concerns listed below:

  1. The author claim that the novelty of this manuscript is, for the first time, to demonstrate the correlation between high NLR and poor OS in pre-treatment and preoperative GBM in a large sample size.  The author needs to discuss more regarding how such a correlation plays a significant role in diagnosis or the following treatment of the gliomas. Most of the studies have shown a correlation between high neutrophil-to-lymphocyte ratio (NLR) and low survival in gliomas no matter before or after treatment. Since the authors didn't draw any conclusions which are different or opposite to the previous studies, the author needs to discuss more the significance of these data to the diagnosis and clinical treatment.
  2. The author mentioned that study conducted by Qian He demonstrated that a high NLR was not associated with poor OS in GBM and the author assumes that the correlation remains controversial. Is this the only study that showed no correlation between NLR and GBM? It is better to discuss the potential possibilities that lead to different conclusions.

Author Response

Dear Reviewer

    Hello

I am Xin Guo, thank you for your valuable suggestions and comments; I will answer your questions one by one and suggest corrections in the draft response letter.

First,  we will discuss further in the discussion section regarding diagnosis and treatment; second, due to an initial translation error, it should have been in the Qian He and Besiroglu et al. study that demonstrated no meaningful relationship between high NLR and low survival in GBM; and finally, we set out in the introduction why we say there is controversy. and mark the modified parts in red.

Finally, thank you again for your valuable comments and I wish you a happy life and good work!

Yours sincerely

Salutations

Reviewer 3 Report

Authors solved all my criticisms.

Author Response

Dear Reviewer

    Hello

     I am Xin Guo, first, thank you for your valuable comments, and I will be more careful and diligent in my future research; second, thank you again for your comments and suggestions, the return draft will be resubmitted on May 12, 2022, as it needs to be re-touched in English language after revision; finally, I wish you a happy life and good work.

Yours sincerely

Salutations
